# Effects of 5-Year Nitrogen Addition on Species Composition and Diversity of an Alpine Steppe Plant Community on Qinghai-Tibetan Plateau

**DOI:** 10.3390/plants11070966

**Published:** 2022-04-01

**Authors:** Ran Zhang, Hao Shen, Shikui Dong, Shuai Li, Jiannan Xiao, Yangliu Zhi, Jing Zhang, Hui Zuo, Shengnan Wu, Zhiyuan Mu, Hang Shi

**Affiliations:** 1School of Grassland Science, Beijing Forestry University, Beijing 100083, China; ran2120921@163.com (R.Z.); abc_zdy@sina.com (H.Z.); 201921180057@mail.bnu.edu.cn (S.W.); shihang03@bjfu.edu.cn (H.S.); 2State Key Joint Laboratory of Environmental Simulation and Pollution Control, School of Environment, Beijing Normal University, Beijing 100875, China; xjn1009@163.com (J.X.); yangliu_zhi@163.com (Y.Z.); jingz121@163.com (J.Z.); wushengnan@163.com (Z.M.); 3Department of Natural Resources, Cornell University, Ithaca, NY 14853, USA; 4College of Resource and Environment, Shanxi Agricultural University, Taigu 030801, China; shuaili@sxau.edu.cn

**Keywords:** N deposition, importance value, species diversity, soil nutrient

## Abstract

The N deposition rate is notably increased in China, especially in the Qinghai-Tibetan Plateau (QTP). How plants respond to the projected N deposition on the alpine steppe is still in debate. In this study, to investigate the effects of N deposition on the plant community of the alpine steppe, we simulated N deposition at six different N addition rate levels (0, 8, 24, 40, 56, 72 kg N ha^−1^ y^−1^) from 2015 to 2019. Species composition and diversity were investigated as the assessment indices. The results showed that the importance value of grasses significantly increased with the increase of the N addition rate, while that of forbs significantly decreased. A high N addition rate (72 kg N ha^−1^ y^−1^) induced species composition change, making *Leymus secalinus* become the most dominant species within the entire plant community. Compared with the control (without N addition), species richness, Shannon–Weiner diversity, Simpson dominance and Pielou Evenness were significantly reduced under a high N addition rate. The changes of plant diversity in the alpine steppe were closely correlated with dynamics of soil nutrients, especially total carbon (TC), total phosphorus (TP) and ammonia nitrogen (NH_4_-N). Our findings suggested that a high N deposition rate (72 kg N ha^−1^ y^−1^) could significantly change plant composition and reduce the diversity of the alpine steppe, though they were less affected by low N deposition rates at present. With the increase of the N deposition rate, plant composition and diversity of the alpine steppe may be negatively affected in the future. In addition, *Leymus secalinus* is more competitive than other species with an N deposition rate increase. Soil C, soil P and soil NH^4^-N variation induced by N deposition might play a key role in regulating changes in plant composition and diversity in the alpine steppe. In addition, longer term field investigation needs to be carried out to testify to this phenomenon with the increase of N deposition in the future.

## 1. Introduction

In the 20th century, the input of N in the global terrestrial ecosystem has continued to increase because of the large-scale burning of fossil fuels, deforestation, and the frequent application of agricultural fertilizers [1,2]. Worldwide, N deposition was about 15 Tg N year^−1^ in 1860, but it jumped to 187 Tg N year^−1^ in 2005 [3]. It is estimated that the global N deposition will reach 200 Tg N year^−1^ by the middle of the 21st century [4]. In China, the total N deposition has increased significantly in the past few decades [5], and its average value was higher than in America and Europe. Although N is the main limiting nutrient element in most terrestrial ecosystems, especially grassland ecosystems [6], increasing N deposition can significantly change soil traits and plant properties, affecting the ecosystem health [7,8]. Therefore, the consequence of increased N deposition has become one of hot issues from the general views of the public, professionals and practitioners.

In recent years, due to the differences of study areas and test methods, the results about the impacts of N deposition on the grassland ecosystem are often inconsistent. Some studies showed that adequate N deposition was beneficial to ecosystems in N-constrained environments [9,10,11]. Other studies indicated that excessive N deposition caused soil acidification and nutrient imbalance, thus affecting the soil and plant health and causing biodiversity loss [12,13,14]. It is important to clarify the responses of grassland plants and soil to the long-term effects of increased N deposition. Species composition is an important indicator to reflect changes in community structure, while plant diversity is an important factor that affects the functional complexity and stability of communities [15]. For grassland communities, numerous scholars have concluded that increased N deposition is one of the important factors affecting species composition and diversity of plant community [16,17,18,19,20,21]. Individual species and functional groups may differ in their response to N deposition because of the variations in N-use strategies and efficiency, causing changes of species composition and losses of biodiversity of the plant community [18,22,23]. For the dominant plant species, a few researchers found that N deposition was beneficial, allowing them to gain more nutrients for rapid growth, thereby reducing the species richness of the community through the loss of some rare species [24]. For the grassland soil, some scholars reported that N deposition changed soil’s physical and chemical properties, thus affecting the growth of individual plants, and composition as well as richness of the plant community [25]. Therefore, it is of great importance to clarify the causes and impacts of increased N deposition on the composition and diversity of plant communities and its relations with soil properties in the grassland ecosystem.

The Qinghai-Tibetan Plateau (QTP), located in southwestern China, is well known as “The Third Pole” of the earth [26]. The total area is 2.5 × 10^8^ km^2^ and the average altitude is more than 4000 m [8]. The alpine steppe is one of the key grassland ecosystems on the QTP [27]. This unique alpine ecosystem in the QTP experiences a significant N deposition of 7.55 kg N ha^−1^ yr^−1^ on average, with a minimum of 1.08 kg N ha^−1^ yr^−1^ and a maximum of 17.81 kg N ha^−1^ yr^−1^ [1,17], and will continuously experience increasing N deposition in the future [4]. Despite numerous studies having indicated that N deposition has a significant impact on plant diversity of the grassland ecosystems in other ecoregions [18,28,29], knowledge regarding the response of the plant species composition and diversity to N deposition remains limited in the alpine grassland on the QTP [27]. On this basis, we conducted this study to explore two issues: (1) the response of species composition and diversity of plant community of the alpine steppe to N addition; (2) the roles of soil factors affecting the plant diversity of the alpine steppe plant community. The results of this study can provide a sound basis for sustainable protection and restoration of the alpine grassland ecosystem on the QTP in the era of increasing N deposition.

## 2. Results

### 2.1. Effects of N Addition on Species Composition of the Plant Community

The NMDS analysis showed that increasing the N addition rate resulted in an obvious shift in species composition along the first axis (Figure 1). The total plant species number/m^2^ (Appendix A) varied with different N addition rates. The total plant species cover (Appendix A) was significantly increased under the high N addition rates.

The effects of the N addition rate on the importance values of component species in the plant community are shown in Figure 2 and Appendix A. The results indicated that the high N addition rate significantly (*p* < 0.05) decreased the importance value of *Aster alpinus* (forb plant). N addition had no significant effect on the importance value of *Carex capillifolia* and *Carex melanantha* (sedge species). The importance value of *Leymus secalinus* (grass species) significantly (*p* < 0.05) increased with the increase of the N addition rate. From the perspective of functional groups, the importance value of grasses significantly increased with the increase of N addition rate, while that of forbs significantly decreased (Figure 3a,b). No significant relationships were found between N addition rate and the importance values in sedges (Figure 3c). 

### 2.2. Effects of N Addition on Species Diversity of Plant Community

The richness index, Shannon–Wiener diversity index, Simpson dominance index, and Pielou evenness index significant decreased with the increase of N addition rate (Figure 4). The effect of N addition rate on plant species diversity is shown in Appendix A. The richness index variation under the N_Ⅰ_ level was insignificantly (*p* > 0.05) higher than that under the control, while the N_Ⅴ_ level led to the lowest richness among all the levels (*p* < 0.05). Compared with CK, the Shannon–Wiener diversity index was significantly (*p* < 0.05) lower than under the N_Ⅴ_ level. The Simpson dominance index under the N_Ⅴ_ level was significantly (*p* < 0.05) lower than that under other levels. The Pielou index was the lowest (*p* < 0.05) under N_Ⅴ_ level.

### 2.3. Relationship between Species Diversity and Soil Nutrients in the Alpine Steppe

Table 1 shows Pearson’s correlations for richness, Shannon–Wiener, Simpson, Pielou, and soil nutrients. Richness was significantly negatively correlated with soil TC (r = −0.471; *p* < 0.05) and NH_4_-N (r = −0.573; *p* < 0.05). There was a strong negatively correlation between Shannon–Wiener diversity and soil TC (r = −0.519; *p* < 0.05), TP (r = −0.532; *p* < 0.05), and NH_4_-N (r = −0.614; *p* < 0.01). There was a strong negatively correlation between Simpson dominance and soil TC (r = −0.591; *p* < 0.01), TP (r = −0.485; *p* < 0.05), and NH_4_-N (r = −0.660; *p* < 0.01). Pielou evenness had a significantly (r = −0.678; *p* < 0.01) negative relation with soil TP. There was no strong correlation (*p* > 0.05) between the plant diversity index and other soil factors. No significant (*p* > 0.05) correlations were found between soil pH and plant diversity index. 

There were significant changes in soil NH_4_-N, TN, TC, TP, AP, AK, and Ca under N addition (*p* < 0.05), while N addition had no significant effect on other soil properties such as, soil NO_3_-N, pH, Mg, and S (Appendix A).

## 3. Discussion

### 3.1. Responses of Plant Species Composition of Alpine Steppe to N Addition

In the present study, the species composition of the plant community of the alpine steppe was altered significantly after 5-year N addition. N addition increased the importance values of grasses. A similar phenomenon was also found in other grassland ecosystems [18,19,20,30]. We also found that the importance value of forbs decreased with increasing N addition rate. These can be explained by the beneficial effects of N addition on the grass group due to their taller individuals and faster growth rate than those of the forb group [19,20]. These responses imply that forbs species may have disadvantages other than grass species in accessing nutrient resources. In other words, N addition improved the growth of grasses species and suppresses the growth of forbs species. In this study, *Leymus secalinus* became the most dominant species in the alpine steppe, suggesting that *Leymus secalinus* was a strong competitor for N resource. Most grass species are nitrophilous and have similar responses [31,32] under N addition. As one representative species of grass, *Leymus secalinus* is a perennial herb with developed rhizomes and it is maybe a kind of nitrophile, making it competitive under the high N load. This might be the good explanation for the shift of plant composition of the alpine steppe into *Leymus secalinus* dominated. All in all, different responses of plant species or functional groups to different levels of N additions led to changes in plant community composition. This implies that the native plant community of the alpine steppe will shift towards another plant community with the increasing N deposition on the QTP.

### 3.2. Responses of Plant Species Diversity of the Alpine Steppe to N Addition

For grassland ecosystems, increased N deposition poses a great threat to biodiversity [17,24]. A large number of N addition experiments had shown that N addition reduced plant diversity in different types of grassland ecosystems, e.g., temperate grassland [29,33,34], acidic grasslands [19,35,36], saline mountain grasslands [28,37]. We found in the present study that the low level of N addition has no effect on species diversity of the alpine steppe, while a high level of N addition can significantly decrease plant diversity of the alpine steppe. This means that the responses of plant species diversity in the grassland ecosystems to N addition may be dependent on the level of N addition. Moreover, this implies that high N addition has a greater impact on the diversity of alpine steppe ecosystems, that is, the higher the N addition, the greater the loss of grassland ecosystem diversity. Additionally, this indicates that N is a key limiting factor for plants growth in the alpine steppe. Possible reasons for declined plant diversity under high-level N deposition might be: (1) the N-preferred plants can be promoted to take more light and water resources via strong competition in the community, e.g., the taller species shaded shorter species, and the dwarf plants were restricted by light and excluded from the community, ultimately leading to a decrease in species diversity [19,38]; (2) increased available N in the soil might accelerate the growth of resource-obtaining species with the faster return on nutrient investment, while it inhibits the growth of resource-conserving species, leading to the declined plant diversity [39]; (3) increased N can promote the density of the litter, which may shade or inhibit the growth of living plants, thereby reducing plant species diversity [18]. 

In summary, the mean annual N deposition is currently 7.55 kg N ha^−1^ yr^−1^ in this study site [1,17], which has very little effect on plant diversity. However, with the continuous projected increase of N deposition in alpine regions [4], the alpine grassland ecosystem is facing the problem of plant diversity decline. Longer-term field investigation is still needed to examine N deposition effects on grassland ecosystem in the future.

### 3.3. Soil Nutrients Affecting Plant Species Diversity of the Alpine Steppe under N Addition

Soil nutrients were the important impacting factors for the plant community composition and biodiversity [40,41]. The effects of soil nutrients on plant diversity were complex. In this study, we found that the plant species diversity was negatively correlated with soil NH_4_-N. Previous scholars [42,43,44] supported this finding by presenting similar results in different research areas. Our previous studies verified that N: P was less than 14 under different N addition treatments in the alpine steppe [17], implying N was still the main limiting nutrient in the alpine steppe. Some studies showed that N deposition can reduce soil N use efficiency by plants in N-limited alpine grasslands [8,41], resulting in the N residue in the soil [45]. The retention of NH_4_-N in the topsoil but not of NO_3_-N may be due to rapid plant uptake, mobility, and lability of any excess nitrate [13,19]. This is why we found a significantly positive relationship between N addition and soil NH_4_-N content in this study. N deposition can make a great deal of NH_4_^+^ accumulation in soil, which promotes some grasses to become dominant plants and restricts the growth of other species, finally leading to the decrease of plant species diversity [19]. We argue that soil is diverse in nature and its complexity makes its interrelationships with biodiversity a challenge for future interpretation [43]. All in all, we found that soil nutrient under N deposition was closely correlated with the species diversity, suggesting that the imbalance of soil nutrient induced by N deposition could be the cause for the change in plant species composition and structure, and the plant diversity as well. 

Appropriate grassland management (e.g., mowing, grazing, and sod cutting) may be an effective way to maintain plant biodiversity under the background of enhanced atmospheric N deposition in the future [46,47,48].

## 4. Materials and Methods

### 4.1. Study Sites

This study was conducted in Tiebujia Village, Gonghe County (99°35′ N, 37°02′ E, 3270 m a.s.l.), which is located at the west sides of Qinghai Lake in Qinghai Province, China [26]. The study site has a typical plateau continental climate with long sunshine hours. Over there, the mean annual temperature is −0.4~1.2 °C, the annual precipitation is 360~430 mm, and the annual evaporation is about 1550 mm [44]. The typical vegetation is the alpine steppe with dominant plant species of *Stipa capillata* L. and *Poa crymophila* Keng. The soil is mostly sandy loam [26].

### 4.2. Experimental Design

In May 2014, 18 plots (replicates) of 2 m × 5 m with 1 m the buffering area were randomly placed in an alpine steppe in Tiebujia Village. The plots were randomly treated by six levels of N deposition, 0 kg N ha^−1^·y^−1^ as the control (CK), 8 kg N ha^−1^·y^−1^ (N_Ⅰ_), 24 kg N ha^−1^·y^−1^ (N_Ⅱ_), 40 kg N ha^−1^·y^−1^ (N_Ⅲ_), 56 kg N ha^−1^·y^−1^ (N_Ⅳ_), 72 kg N ha^−1^·y^−1^ (N_Ⅴ_), with three replicates for each N treatment (Appendix A). According to Zong et al., (2016) [49], 50 kg N ha^−1^·y^−1^ critical loads, the high N addition level mainly is intended to simulate the N saturation state. All the treatments had similar topographic conditions and land use histories. The N deposition was simulated with fertilization with granular ammonium nitrate (NH_4_NO_3_) in early May and July every year since 2014. Until 2019, we have fertilized for 5 years continuously. 

### 4.3. Plant and Soil Sampling

In July 2019, we randomly placed one 1 m × 1 m quadrat in each plot for investigating plant species composition including name, abundance, coverage, and frequency of each species. Species abundance is the number of species in the community. Species coverage refers to the percentage of the vertical projected area of the above-ground part of the species to the surveyed area. Species frequency refers to the frequency of occurrence of the species within the survey range. After recording the plant species composition, we collected the soil samples at a depth of 0–20 cm depth soil cores from each plot. The collected soil samples were sealed in Polyethylene bags and transported to the lab for analysis after being air-dried, ground, and sieved (a 100 mesh sieve was used for soil nutrients measurement and 18 mesh for pH measurement).

An elemental analyzer (Elementar, EA 3000, Langenselbold, Germany) was applied to measure the total nitrogen (TN) and total carbon (TC) content of the soil. Ammonium nitrogen (NH_4_-N) and nitrate nitrogen (NO_3_-N) were measured using a flow injection auto-analyzer (Tianjin Zhongtong Technology Development, AACE, Berlin, Germany). We used ICP-MS (SPECTRO ARCOS EOP, Germany) to measure the total phosphorus (TP) according to the following procedures. Soil samples were first digested in sulfuric acid and then quantified using the ICP Elemental Analyzer [17]. The concentrations of available phosphorus (AP) were extracted from soil samples with calcium chloride (CaCl_2_) solution, and the supernatant was collected with a pipette and stored at 6 °C until analysis [17]. The concentrations of available potassium (AK) are measured similarly to the available phosphorus (AP), but Ammonium Acetate solution is used to extract the available potassium (AK) from soil samples. AP and AK were measured by ICP-MS (SPECTRO ARCOS EOP, Kleve, Germany) [17]. Soil pH was measured with a pH meter using a 1:2.5 soil: water ratio [50]. Calcium (Ca), Magnesium (Mg), potassium (K), and sulfur (S) were measured by atomic spectrometry (AA-610S; Shimadzu, Kyoto, Japan) [8].

### 4.4. Statistical Analysis

#### 4.4.1. Species Importance Value

We calculated the importance value (IV) of the plant species using the following formula [31]:IV = (relative frequency + relative coverage + relative abundance)/3 × 100%(1)

The relative frequency is the percentage of the frequency of individual species in the sum of the frequencies of all species. The relative coverage is the percentage of the coverage of individual species in the sum of coverage of all species. The relative abundance is the percentage of the number of individuals species in a unit area to the number of all species in a unit area.

#### 4.4.2. Species Diversity Index

Plant species diversity was evaluated using the four-dimensional biodiversity indices, i.e., richness index (R), Shannon–Weiner diversity index (H), Simpson dominance index (D), Pielou Evenness index (J), which were calculated based on the following formulas [30,43,51]:R = S(2)
where R represents richness index, S means the total number of species in the community.
H = −∑P_i_ lnP_i_(3)
where H represents Shannon–Weiner diversity index, P_i_ is the ratio of the number of individuals belonging to species i to the total number of individuals in the community.
D = ∑P_i_^2^(4)
where D represents Simpson dominance index, and P_i_ is the ratio of the number of individuals belonging to species i to the total number of individuals in the community.
J = H/lnS(5)
where J represents Pielou Evenness index, H represents Shannon–Weiner index, S is the total number of species in the community.

#### 4.4.3. Statistical Analysis

Our data were presented as means ± standard error (SE) in figures and tables. The ‘vegan’ and ‘ggplot’ packages in R (4.1.2) were used for Nonmetric multidimensional scaling (NMDS) of species abundance among different N addition levels. Duncan’s Multiple Range Test in analysis of variance (ANOVA) was used to test the effects of different N addition on the plant community composition, diversity indices, and soil nutrients in SPSS 23.0. We employed the simple linear regression model to determine the relationship between functional group importance value, plant diversity, and N addition rate. We used SPSS 23.0 to perform Normality Test and assessed the relationship between plant diversity and soil nutrients by using Pearson‘s correlation. GraphPad Prism 8.0 was used for drawing.

## 5. Conclusions

From this 5-years N addition study, we found that importance values of different functional groups and species responded differently to N addition. It is clear that the importance value of the grass functional group significantly increased, while that of forbs significantly decreased with the increase of N addition. *Leymus secalinus* was the most adaptive species within the entire community. The high level of N addition (72 kg N ha^−1^·y^−1^) has a greater impact on the diversity of alpine steppe ecosystems. Additionally, under the continuous N addition for 5 years, the changes of plant diversity in the alpine steppe had a close relation with soil nutrients, especially total carbon (TC), total phosphorus (TP), and ammonia nitrogen (NH_4_-N). Soil nutrients imbalance induced by long term N addition may be the main cause that leads to the species diversity loss and composition shift in alpine steppe.

## Figures and Tables

**Figure 1 plants-11-00966-f001:**
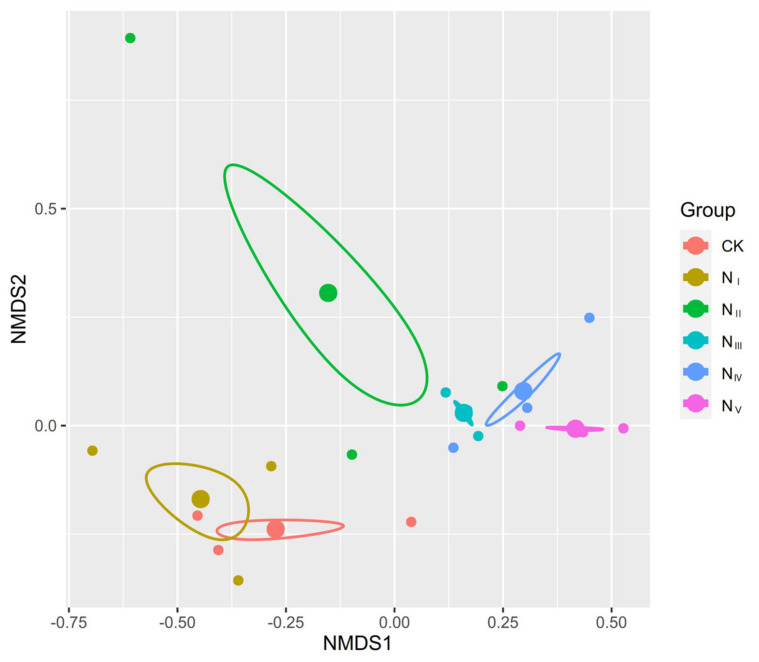
Non-metric multidimensional scaling plot of species abundance between different N addition rates. Dots of the same color come from the same group. Each small dot represents a plot replication, and each large dot represents the mean value of three plot replications. Circles indicate the 95% confidence of the mean value. The closer the distance between the two points or circles, the smaller the difference between the two groups. Abbreviations: CK, N_I_, N_II_, N_III_, N_IV_, and N_V_ are 0, 8, 24, 40, 56, and 72 kg N ha^−1^·y^−1^, respectively; similarly for the following figures and tables.

**Figure 2 plants-11-00966-f002:**
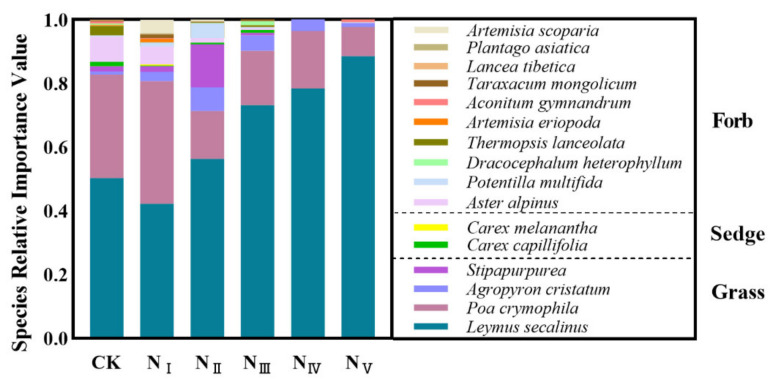
Species composition and importance values of plant communities of the alpine steppe under different N addition rates.

**Figure 3 plants-11-00966-f003:**
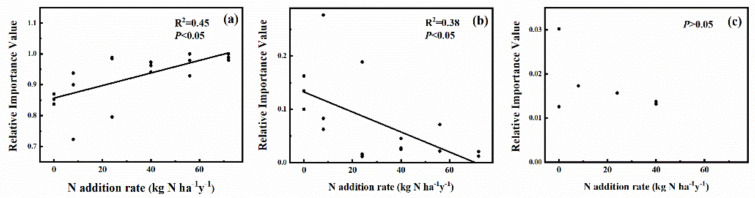
Relationship between functional group importance values and N addition rates: (**a**) grass functional group, (**b**) forb functional group, (**c**) sedge functional group.

**Figure 4 plants-11-00966-f004:**
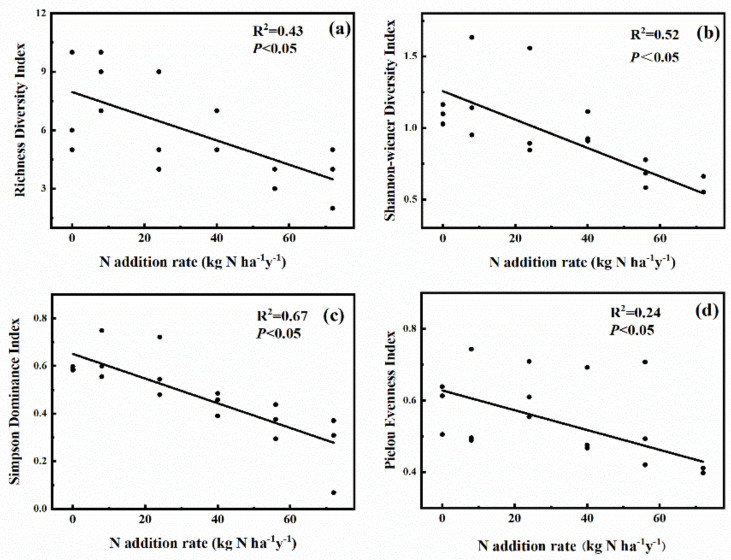
Relationship of Richness index: (**a**), Shannon–Weiner index (**b**), Simpson index (**c**), Pielou evenness index, (**d**) with N addition rate.

**Table 1 plants-11-00966-t001:** Person’s correlation test between species diversity and soil nutrients.

Variables		Plant Diversity			
		R	H	D	J
Soil	TN	−0.217	−0.229	−0.338	−0.13
	TC	−0.471 *	−0.519*	−0.591 **	−0.427
	TP	−0.158	−0.532*	−0.485 *	−0.678 **
	TK	0.066	−0.017	−0.017	−0.089
	NO_3_-N	−0.214	−0.275	−0.201	−0.242
	NH_4_-N	−0.573 *	−0.614 **	−0.660 **	−0.435
	AP	0.027	−0.14	−0.118	−0.257
	AK	0.224	0.156	0.207	0.059
	pH	0.133	−0.121	−0.1	−0.308
	Ca	−0.425	−0.33	−0.462	−0.04

Note: * Indicates the significance level *p* < 0.05; ** indicates the significance level *p* < 0.01. Abbreviations: R is Richness Index; H is Shannon–Wiener diversity Index; D is Simpson dominance Index; J is Pielou evenness Index; TN is total nitrogen content; TC is total carbon content; TP is total phosphorus content; TK is total potassium content; NH_4_-N is ammonium nitrogen content; NO_3_-N is nitrate nitrogen content; AP is available phosphorus content; AK is available potassium content; Ca is Calcium content; Mg is Magnesium content; S is sulfur content.

## Data Availability

All the data provided in this study are available within this article.

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
