# Peer review of "Effects of 5-Year Nitrogen Addition on Species Composition and Diversity of an Alpine Steppe Plant Community on Qinghai-Tibetan Plateau"

_plants, 2022, doi:10.3390/plants11070966_

Round 1
Reviewer 1 Report
The authors carried out an interesting experiment on the Tibetan Plateau, concerning the effects of Nitrogen addition on grassland ecosystem. Particularly, they tested six different levels of N addition (simulating increasing depositions) on plant species composition. The study is sound and of relevant interest, in the context of the increasing global pollution, aiming at filling a gap of knowledge about the Tibetan Plateau area.
Despite this, there are some severe limitations, particularly concerning the statistical analyses, which appears, in my opinion, not appropriate at all. Indeed, some multivariate approach would be more suitable for this kind of studies. For instance, I would have expected an Indicator Species Analysis or a Permanova, with Simper analysis to detect relevant difference in species composition. Another relevant gap in this paper is the lack of information about total species cover. Did N addition improve total grassland cover? This would be interesting for soil conservation and for management purposes. Furtherly, three replicates for each treatment are a very limited number of replicates. I wonder if this limitation would definitively hamper the publication of the study on Plants journal.
Additionally, the authors do not provide a clear message, with some practical implications, even if they speak about ‘management strategies’.
To improve the readability of the paper, I also suggest replacing ‘nitrogen’ with ‘N’ throughout the manuscript.
Minor and additional comments:
ABSTRACT
The experimental design is not clear. You have to specify which treatments you applied before reporting the results in this abstract.
L20 What is the ‘important value’? provide a short description
L27 no need to repeat 72 kg ha y, after you have specified that it belong to the ‘high N addition’
It is not clear what is your conclusion: which is the ‘best’ treatment? If there is one
KEYWORDS
These keywords are all already written in the title. You should find other words not already mentioned in the title
INTRODUCTION
L 57-58 plants, not plant; additionally, N deposition is a natural phenomenon, maybe you would refer here to ‘increased N deposition’?
L58-67 These three sentences seem a little bit repetitive among each other. Try to simplify and harmonize this concept.
L67 what is a ‘dominate plant species’? Maybe you would mean ‘dominant plant species’?
L70 ‘that that’ I think is a typo.
L86-88 since you applied six different levels of N additions, this should be reported in the abstract
RESULTS
L95-101 there is no statistical evidence of the significant effect on species variations. mongolincum --> mongolicum
L104-106 you have to indicate either P≤0.05 or P≥0.05
L120 what is a significant effect with P>0.05?????
L131 Realationship --> Relationship; diveristy --> diversity
L131-L140 provide R2 values for the correlation coefficients reported in the text.
FIGURES
Figure 1: Potentilla multifidi --> Potentilla multifida
Figure 2: Since the other figures are presented with treatments on X axis and the teste variable on Y axis, I would expect to see the same here.
Figure 3: there is a mistake in the caption, please correct Figure 2 --> Figure 3. Additionally, there is no reason to present the results concerning the different N levels with different shades. Just use one colour (possibly not black, which hampers to see error bars).
Figure 4: I cannot understand why you presented all the possible correlations, while there are some which are completely non-sense, such as R-H-D-J among each other. I think there are better ways to present such results, not strictly in a colour image but rather maybe in a table.
DISCUSSION
L164 dominate --> dominant
L167 how can you say that Leymus is a nitrophilous plant? Do you have any reference to support this? competative --> competitive
L168 nitrophilo --> nitrophilous
L173 I am not sure you can refer to ‘secondary grasslands’ in this case
L175 apline --> alpine
MATERIALS AND METHODS
L221-227 is there any herbivore grazing these grasslands? Or any other active grassland management?
L222 ASL --> a.s.l.
L224 which average temperature? Annual? To which time period refers to?
L226 please add authors after species names for a scientific notation
L228 how far were the plots each other? Were they in similar topographic conditions? And what about species compositions? The homogeneity among plots should be demonstrated before carrying out the subsequent analyses (e.g. through a Permanova on species coverages or something else).
L229 check English of this sentence
L235-237 maybe I did not understand properly, but it seems to me that these two last sentences report the same information (the second one can be delated). I would be glad to read ‘for five years consequently’ in this description.
L230-233 6 levels and 3 replicates should be in letters not in numbers
L240 which is the difference among abundance, coverage and frequency of the species you recorded? I can find the description in L260-263, but it should be replaced here. Anyway, I cannot understand the difference between species frequency and species density… And you should explain how you assessed the species coverage (i.e. the reference methodology)
L259 overage --> coverage. Additionally: is there any reference for this formula? Or did you experiment this for the first time?
L265-281 please provide references for these formulas
L284 imprtance --> importance
L283-285 did you tested for normality and homoscedasticity of Anova residuals? Why did you not apply the REGWQ or Tukey post-hoc test (which should be more appropriate for such balanced experiments)?
L285-287 did you test for normality the response variable? If this assumption was not meet, Spearman coefficient should be considered instead of Pearson. Moreover, why did you not apply a regressive model also including the N addition level as fixed factor? Did you expect a causal relation between the couples of variables? If yes (as long I can understand), a regression (even just simple linear ones) should be applied. Additionally, why did you use N addition as a continuous variable (in such correlations) instead of a 6-levels categorical variable? There is no reason to change the approach applied while setting the experiment.
CONCLUSION
L296-299 which kind of management can you suggest? There should be some solution able to reduce or remove N deposition to protect grasslands against species loss… please report them here (or in the Discussion section), if any.
ACKNOWLEDGMENTS
L305-306 fifinancially --> financially; Scientifific --> Scientific
L309-311 remove these sentences
Reviewer 2 Report
There is a long history of examining the effects of nitrogen on grassland communities. While examining these effects in a high elevation community has some potential interest, I was disappointed by several aspects of this study. The basic experimental setup is OK, but the level of replication is low (3 plots per nutrient level). Additionally, a small subsection of the plots was examined and only at one time period. These low levels of replication and sampling may explain the lack of an effect of N on the forbs. I think it would have been better to do more sampling per plot, have more plots and sample them more frequently.
The basic result that high levels of N increased the abundance of grass is well known. It is perhaps more surprising that lower levels of N addition had no effect on the community.
Repeatedly the authors make statements about their data that are not correct. For example, there is a statement that low levels of N addition increased diversity. This did not occur.
I sympathize with the difficulty of writing a paper in English, but there were too many mistakes in the manuscript for it to be understood. I am attaching a Word file with suggested edits. These are not a complete list of required changes.

Reviewer 3 Report
The manuscript entitled ”Effects of 5-year nitrogen addition on species composition and diversity of Alpine Steppe plant community on Qinghai-Tibetan Plateau" is interesting and acceptable for Plants after the following revisions.
Abstract: It is necessary to add the reason for increasing total phosphorus by high nitrogen deposition because total P in the soil will not increase unless P was added by some sources.
Introduction: The authors used the Patrick richness index, Shannon-Wiener diversity index, Simpson dominance index, and Pielou evenness index for the evaluation of plant species diversity. Please explain what is the characteristics and advantages of these indexes.
Results:
Figure 1: The authors show the percentage composition of plant species, but the number of species may be also important. Please add the changes of total plant number /m2.
Figure 2: The authors divided the graph into three separate parts because the Relative Importance values of grasses, forbs, and sedges are significantly different. I suggest making three separate graphs for grasses, forbs, and sedges.
Results: The data for soil analysis is not shown. Please add the data of soil analysis in the text or supplemental tables.
Materials and Methods:
Line 235-236: The last sentence is the repetition of the former sentence.
Line 547-248: Please add the sample preparation of the soil before determining the total P content.
Round 2
Reviewer 1 Report
The authors responded to all the points I addressed in the previous revision form. In my opinion the manuscript can be published in the present form.
I just have two small remarks, concerning the supporting information file:
- in S2 to S5 tables provide a reasonable and even number of decimal digits (e.g. 83.00 instead of 83 in S1 or 0.50 instead of 0.50291978 in S3)
- the values reported in S4 are R and not R2 values, being the result of a correlation (this is my blame, I suggested you to add R2 values in the previous revision, instead of R values)
Reviewer 2 Report
This version of the manuscript is an improvement over the previous version. However, given the limitations of the replication and the sampling, I would not rate this as a very important contribution. The authors have added a multivariate analysis to the results. What methods were used to generate this analysis are not adequately described, most especially, what data was used for the analysis. While I think there are some technical issues with the presentation, I think it does help to summarize the change in the community over the N addition gradient.
As with the previous version, the data show that when the N addition treatments are compared, there is very little effect on the vegetation until the highest level of N addition. These high levels of N addition are well above the level of atmospheric deposition for the region. This is never discussed in the manuscript. I think it should be emphasized so that the reader is not under the impression that the present rates of N deposition are having an impact on the vegetation. I also think it would be better to present the results using the N addition as a continuous variable. This is done in the supplementary material but is really a more powerful way to analyze the data.
There are lots of minor changes that I have listed in the attached document. So, I think the manuscript has a lot of work still, but can be acceptable for publication one the changes are made.

Author Response
请看附件
